# Genome-Wide Association Mapping of Grain Metal Accumulation in Wheat

**DOI:** 10.3390/genes13061052

**Published:** 2022-06-13

**Authors:** Mohamed El-Soda, Maha Aljabri

**Affiliations:** 1Department of Genetics, Faculty of Agriculture, Cairo University, Giza 12613, Egypt; 2Department of Biology, Faculty of Applied Science, Umm Al-Qura University, Makkah 24231, Saudi Arabia; myjabri@uqu.edu.sa

**Keywords:** biofortification, GWAS, metals, superior genotypes, molecular marker

## Abstract

Increasing wheat grain yield while ignoring grain quality and metal accumulation can result in metal deficiencies, particularly in countries where bread wheat accounts for the majority of daily dietary regimes. When the accumulation level exceeds a certain threshold, it becomes toxic and causes various diseases. Biofortification is an effective method of ensuring nutritional security. We screened 200 spring wheat advanced lines from the wheat association mapping initiative for Mn, Fe, Cu, Zn, Ni, and Cd concentrations. Interestingly, high-yielding genotypes had high essential metals, such as Mn, Fe, Cu, and Zn, but low levels of toxic metals, such as Ni and Cd. Positive correlations were found between all metals except Ni and Cd, where no correlation was found. We identified 142 significant SNPs, 26 of which had possible pleiotropic effects on two or more metals. Several QTLs co-located with previously mapped QTL for the same or other metals, whereas others were new. Our findings contribute to wheat genetic biofortification through marker-assisted selection, ensuring nutritional security in the long run.

## 1. Introduction

Wheat (*Triticum aestivum* L.) is one of the world’s most important crops and a staple food in many developing countries. It plays a vital role in human health by providing carbohydrates, proteins, and certain inorganic metals. However, due to the increase in the world’s population, most wheat research programmes focus on increasing yields, ignoring grain quality, leading to nutrient deficiencies, especially in countries where bread wheat forms the majority of daily calories intake [1]. Although the daily requirements for inorganic metals, such as manganese (Mn), iron (Fe), copper (Cu), and zinc (Zn), are as low as a few mg, Fe and Zn deficiencies are common in developing communities that depend mainly on wheat in their dietary regime. However, metals become toxic at high concentrations [2,3,4,5]. Nickel (Ni) is another metal required by plants in trace amounts. Although Ni does not have any special metabolic functions, it is a component of several enzymes, such as ureases and hydrogenases. Its absence from the soil decreases urease activity and disturbs nitrogen assimilation. In addition, Ni is necessary for cereals growth and grain production [6]. Cadmium (Cd) is a highly toxic heavy metal with no known beneficial effects on plants. It is chemically similar to Zn and can compete for common mechanisms for uptake and translocation in the crop [5,7]. In general, the highest accumulation levels of Ni and Cd are in roots, then in leaves, and the lowest in seeds [4].

In general, wheat grain consumption is safe when the accumulation of the metals is under the accepted safety thresholds. However, when it exceeds this threshold, it may have toxic effects and cause various diseases in humans [4,8]. The accepted safety thresholds in mg kg^−1^ are 500 for Mn, 73.3 for Cu, 99.4 for Zn, 67.9 for Ni, and 0.2 for Cd [4]. In contrast, to combat essential metal malnutrition, biofortification was proposed as a complementary strategy where stable genotypes with high micronutrient accumulation have been characterised and selected for future crosses through traditional or modern plant breeding techniques [9].

Such biochemical traits are known as quantitative traits influenced by the environment, and controlled by quantitative trait loci (QTL) [10]. Genome-wide association mapping (GWAM) is a common approach to dissecting complex phenotypes and mapping the associated markers. GWAM uses linkage disequilibrium between polymorphic molecular markers and the causal gene. This approach depends on large panels of breeding lines or genotypes collected from naturally evolved and adapted populations with wider genetic variation. It can often identify smaller intervals by making elegant use of historical recombination events using polymorphic markers, such as single nucleotide polymorphisms (SNPs). As a result, dense maps and high statistical mapping resolution facilitate identifying SNPs associated with the studied trait [11,12,13,14]. These associations and candidate genes may provide key markers for trait introgression, marker-assisted selection, or targets for functional manipulation for crop improvement.

Another approach that has gained popularity in plant breeding research is genomic prediction (GP), which depends on using molecular markers that cover the entire genome. This approach uses genome-wide marker information to predict the breeding value of complex traits to speed up breeding programmes [15]. A critical method for GP is ridge regression–best linear unbiased prediction (rrBLUP) that allows for efficient prediction with unreplicated training data [16].

We used 200 spring wheat advanced lines to investigate the accumulation of six metals in wheat grains and select genotypes with the desired metal accumulations, to map the associated SNPs.

## 2. Material and Methods

### 2.1. Plant Material, Metal Measurements, and Statistical Analysis

We used 200 spring wheat advanced lines from the wheat association mapping initiative (WAMI) population [17], genotyped with 26,814 SNPs [18] and released by the International Maize and Wheat Improvement Center (CIMMYT). The soil texture was sandy clay loam, from 0 to 30 cm, followed by sandy loam soil, from 30 to 45 cm. The experimental design was a randomised block design with three replicates. Each genotype was represented in each block as a row of 2 m and a spacing of 0.10 m between plants. The experiment received 238 kg ha^−1^ ammonium nitrate (33.5% N), 75 kg ha^−1^ calcium superphosphate (15.5% P_2_O), and 58 kg ha^−1^ potassium sulfate (48% K_2_O).

A concentrated HNO_3_ and HCLO_4_ (10:4) mixture was added to 0.5 g of crushed seeds in the digestion vessel, then closed and heated in a water bath at 80 °C for 120 min. The solution was cooled to room temperature and filtered through Whatman No. 1 filter paper into 50 mL in a volumetric flask with double-distilled deionised water. An atomic absorption spectrophotometer (AAS) was used to determine the concentrations of Mn, Fe, Cu, Zn, Ni, and Cd in the extract.

All statistical analysis was run using the raw data in R for Windows 4.1.2, and for drawing frequency distributions, we used the R package corrplot.

### 2.2. Genome-Wide Association Mapping and Genomic Prediction

The TASSEL software, version 5.0 [19], assisted in identifying SNP markers associated with the raw data of the measured metals using the mixed linear model (MLM), kinship matrix, and principle component.

The software package for R called ridge regression–best linear unbiased prediction (rrBLUP) [16] was used to evaluate the genomic predictions. We applied four- and five-fold cross-validations to the 200 examined genotypes and the training sets comprised 140 and 158 randomly chosen genotypes, respectively. The correlation between the observed and predicted values supported the calculation of the prediction ability, and then the mean correlation accuracy was calculated after 100 iterations.

### 2.3. Candidate Genes Identification

We obtained the flanking sequences for the significant SNPs [20] and then identified candidate genes using the BLAST service available at the GrainGenes (https://wheat.pw.usda.gov/blast/, accessed on 15 March 2022) and the wheat IWGSC RefSeq v2.1 [21]. We used the KnetMiner gene discovery platform [22] (https://knetminer.org, accessed on 15 March 2022) to search for large-genome-scale knowledge graphs and visualize interesting subgraphs of the related information about the biology and functions of genes, gene networks, and traits.

## 3. Results

### 3.1. Statistical Analysis

A wide range of genetic variations was observed for all traits (Table 1). The lowest observed variation was for Mn, ranging from 3.0 to 9.9 mg kg^−1^ with an average of 6.1 mg kg^−1^. Zinc had the highest variation, ranging from 9.9 to 88.8 mg kg^−1^. Iron and Ni showed moderate variations with minimum values of 5.3 and 11.1 mg kg^−1^, respectively, and maximum values of 19.2 and 31.0 mg kg^−1^, respectively. The frequency distribution (Figure 1) showed normal distribution for all traits.

Five genotypes selected for their high grain yield [23] showed high Fe and Zn and low Ni and Cd contents (Table 2). Another eight genotypes showed the highest Fe and Zn content, and their Cd content was low to moderate. An additional 18 genotypes were higher than the accepted Cd safety threshold.

Apart from the 18 genotypes with high Cd content, our results showed that, on average, all metals reported here were lower than the accepted safety thresholds.

The correlation plot (Figure 2) showed a positive correlation between all measured traits, except Ni and Cd. The highest correlation was 0.70 between Mn and Fe, followed by 0.40 between Fe and Cu. The correlation between Fe and Cd was 0.32, and the correlation between Cu and Cd was 0.23. The lowest correlation, 0.13, was observed between Cu and Ni and between Zn and Cd.

### 3.2. Genome-Wide Association Mapping

One hundred thirty-eight significant SNPs, above –log10 (*P*) = 3, were associated with the measured metals (Figure 3, Table 3, and Appendix A). For example, one QTL associated with Mn harboured 25 SNPs and 12 genes on 6A at 48 cM, of which 10 and 15 SNPs decreased and increased Mn content, respectively. One more QTL associated with Fe sheltered nine SNPs located in five genes on 5B at 60 cM. Four and five SNPs showed negative and positive effects, respectively. The nine SNPs had A as the major allele and C as the alternate allele.

Interestingly, a QTL on 5B, located between 212 and 215 cM, was associated with Mn and Fe. This QTL harboured five SNPs. The same allele of each SNP affected both traits in the same direction; however, the effect on Fe was always double the effect on Mn. Three of those SNPs had positive effects, and two SNPs showed adverse effects on both traits. Two additional QTLs on 6A and 6B decreased the Cu and Cd content. Both of these QTL showed less significant associations with decreasing Fe content (2.7 and 2.8, respectively). The QTL on 6A was located between 125 and 126 cM, and the SNP was mapped to TraesCS6A03G0953900.

### 3.3. Genomic Prediction (GP)

We expanded our study to include the GP, and the predictability values for our wheat panel were low to moderate, ranging from 0.15 to 0.65. Using a training population of 140 genotypes revealed a prediction of 0.34 for Mn, 0.39 for Fe, 0.21 for Cu, 0.14 for Zn, 0.17 for Ni, and 0.20 for Cd. Increasing the size of the training population to 158 showed similar prediction values, namely, 0.32, 0.37, 0.19, 0.15, 0.15, and 0.21 for Mn, Fe, Cu, Zn, Ni, and Cd, respectively.

## 4. Discussion

Wheat grains are inherently low in essential metals, such as Mn, Fe, Cu, and Zn, which leads to metal malnutrition: a problem affecting a large number of the human population that depends on wheat as a staple crop. Manganese and Cu deficiencies in humans are not dangerous, but together with Fe and Zn, they play central roles in growth, development, and the immune system. Therefore, there is a higher priority and a fundamental need to grow healthy wheat with high concentrations of essential metals and low concentrations of toxic metals for humans, such as Ni and Cd.

Biofortification is a sustainable and cost-effective solution that relies on selecting genotypes with accepted levels of metal accumulation and dissecting the genetic architecture underlying their natural variation [8,9,24,25,26]. We selected the best performing genotypes based on different criteria. First, we reported the metal content of the five highest yielding genotypes that overlapped with an earlier study [23] that evaluated the same populations under the same environmental conditions. Interestingly, those genotypes were among the highest in Mn, Fe, Cu, and Zn, and the lowest in Ni and Cd content, supporting their suitability for future breeding programmes. The second group included eight genotypes with high Fe and Zn content, recommended by an introgression scheme to introgress genes underlying high Fe and Zn content into high-yielding genotypes.

The metal mean values reported here are comparable with earlier reports for winter wheat. For example, the mean values for Mn and Fe, 6.1 and 12.6 mg kg^−1^, respectively, are comparable with previous measurements of 8 and 13 mg kg^−1^ [27], and 10 and 12 mg kg^−1^ [28], respectively. The concentrations of Fe, Cu, and Zn reported here agree with a report on Iranian wheat [2], and the Zn and Cd concentrations agree with reports on Chinese wheat [24,28,29].

The positive correlation observed here between Mn, Fe, Cu, and Zn is similar to earlier observations for synthetic hexaploid wheat [8], Chinese wheat [28], and European elite registered varieties [1]. The positive correlations between Cd, Mn, Fe, and Zn align with an earlier report [7] where high-Cd accumulating wheat varieties had high Ca, Mg, Mn, and Fe. In contrast, low-Cd accumulating wheat varieties were deficient in essential nutrients such as Zn. Therefore, reducing the Cd allocation in wheat is the ultimate target of breeding programmes. However, this is a challenging task as Cd transporters are also transporters of essential metals, such as Fe, Zn, or Mn [30]. One way to overcome this constraint could be the introgression of essential metals genotypes with low Cd accumulation. A recent study reported the introgression of Zn in wheat as a step toward mitigating malnutrition among the masses in developing countries [31].

The highest correlation observed here was 0.70 between Mn and Fe, for which we mapped a QTL on chromosome 5B harbouring five significant SNPs located between 708,652,692 and 708,655,527 bp in TraesCS5B03G1356900. Three significant SNPs in the same gene were reported recently to be associated with Cu [28]. The alternative name of this gene is *PGR5-LIKE A,* and in Arabidopsis, At5g59400; *AtPGR5LA* is tightly co-expressed with several other genes—*FER1*, *FER4*, *NAP1*, *NAS3*, and *YSL1*—that are involved in ROS detoxification and iron distribution [32]. Two more SNPs in TraesCS1B03G1211300 were associated with Mn and mapped to chromosome 1B at 141 cM, between 677,600,765 and 677,607,233 bp, co-located with a significant SNP associated with Mn and Zn at 142 cM, between 676,697,120 and 676,706,013 [28]. On chromosome 2B, we found three significant SNPs associated with Fe at 99 cM co-locating with three significant SNPs associated with Cu at 108–109 cM and one significant SNP associated with Zn and Mn [28]. We mapped two significant SNPs on chromosome 5B at 96 cM associated with Cu co-located with SNPs associated with Cu, Zn and Mn [26]. Our QTL on chromosome 3A was associated with Ni and harboured 40 significant SNPs, of which 33 SNPs were between 85 and 86 cM, and 7 SNPs were at 93 cM, co-located with a previous QTL associated with Cu, Fe, S, and P. Another 25 significant SNPs associated with Mn on chromosome 6A at 48 cM co-located with Fe, S, and Cu [33]. The SNP marker Excalibur_c92298_213, located between 498,311,693 and 498,314,962 bp on chromosome 1D, was associated with Mn and Fe with −log10(*P*) values of 3.4 and 2.7, respectively. This SNP was very close to a significant SNP associated with Cd located between 493,863,084 and 493,875,788 [28]. Several co-locations were observed for the correlated metals Cu and Cd (0.23) on chromosomes 6A (between 600,667,268 and 600,672,612 bp) and 6B (between 695,976,232–695,981,168 bp). However, these two QTLs are not favourable for breeding programmes as they reduce both metals and their effect on Cu is 14 times the effect on Cd. The observed co-locations are not unique to our study but were observed recently in wheat [26]. Altogether, our results indicate pleiotropic effects of the significant SNPs on the measured traits.

In general, GP is a promising approach for enhancing complex traits, such as metal content in wheat, primarily when a large germplasm panel with high numbers of markers is investigated. Both requirements facilitate more accurate estimates of breeding values that were low to moderate for the measured metals, which is in agreement with previous results for wheat [1,34,35].

## 5. Conclusions

Upon evaluating 200 wheat genotypes for six metals, we selected five high-yielding genotypes with the desired concentrations of the measured metals. In addition, we selected eight genotypes with the highest Fe and Zn content for possible use in future breeding programs. In addition, we mapped 138 SNPs with favourable allelic effects essential for selecting genotypes with high micronutrient content via marker-assisted breeding programs. The identified candidate genes need to be confirmed or modified through genome editing technology.

## Figures and Tables

**Figure 1 genes-13-01052-f001:**
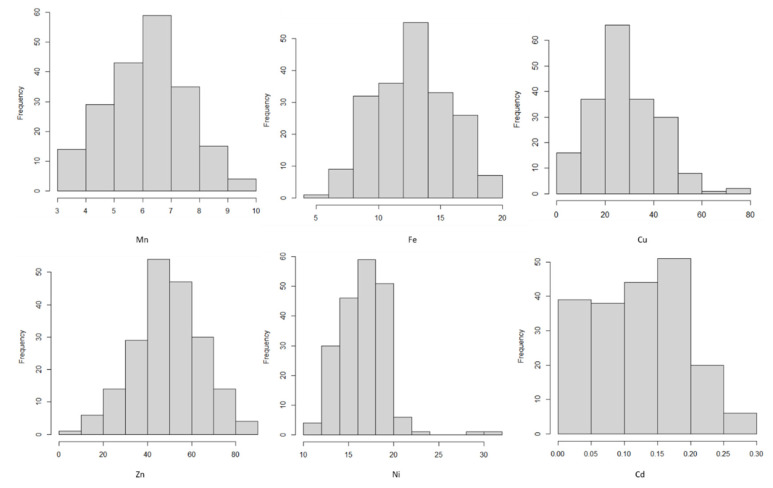
Frequency distributions of non-normalised metals values for the WAMI population. Vertical axes indicate the number of lines per trait value class, and horizontal axes indicate the different trait value classes.

**Figure 2 genes-13-01052-f002:**
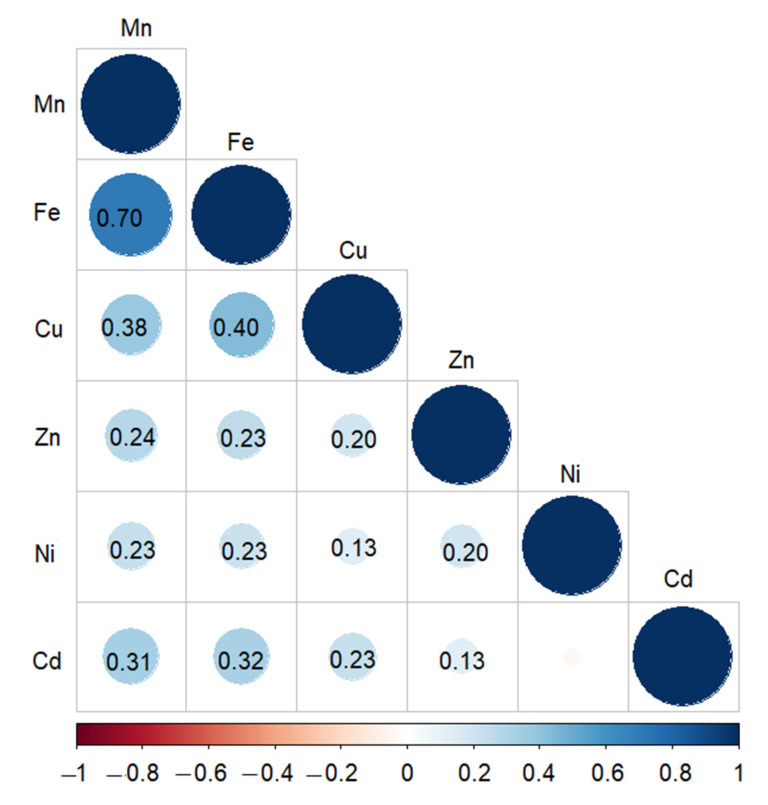
Correlation plot for the measured metals. The size and the colour of the circles indicate the significance level. The blue colour indicates a positive correlation between the measured metals.

**Figure 3 genes-13-01052-f003:**
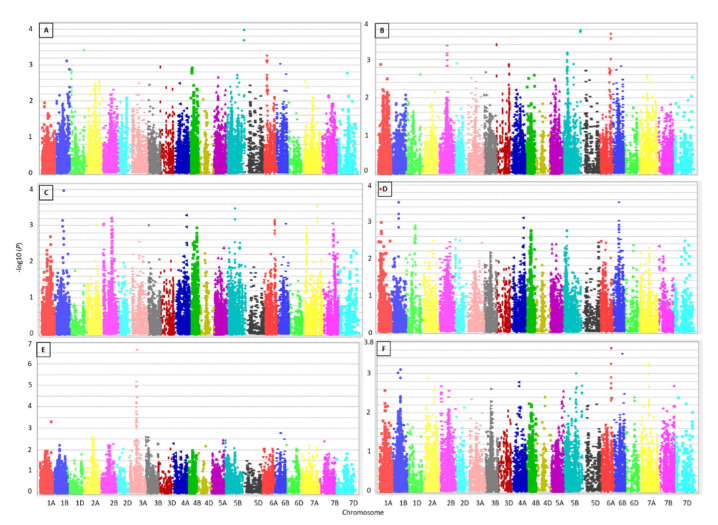
Manhattan plots representing association mapping for the 200 wheat genotypes using 26,814 SNPs for Mn (**A**), Fe (**B**), Cu (**C**), Zn (**D**), Ni (**E**), Cd (**F**). The horizontal axis represents the 21 wheat chromosomes, and the vertical axis represents the significance level −log10 (*P*) values.

**Table 1 genes-13-01052-t001:** Population performance for the measured metals. Minimum (min), maximum (max), and mean values in mg kg^−1^. S.E. represents standard error values. Accepted safety level threshold according to Al-Othman et al., 2016 [4].

Metal	Min	Max	Mean	S.E.	Safety Level
Mn	3.000	9.930	6.100	0.099	500
Fe	5.299	19.230	12.633	0.213	-
Cu	4.200	77.300	28.247	1.093	73.3
Zn	9.900	88.800	48.949	1.077	99.4
Ni	11.100	31.000	16.596	0.191	67.9
Cd	0.003	0.266	0.123	0.005	0.2

**Table 2 genes-13-01052-t002:** Metals content (mg kg^−1^) for genotypes with the highest yield, the highest Fe and Zn content, and the highest Cd content.

Genotype	Mn	Fe	Cu	Zn	Ni	Cd	Category
393392	7.00	17.81	48.70	39.50	15.60	0.16	High yield
1706327	7.82	16.85	30.60	48.70	12.00	0.08	
346403	7.90	15.80	24.00	29.80	15.20	0.21	
3597332	9.15	13.12	78.50	52.50	18.40	0.06	
294568	5.10	9.81	11.50	50.70	15.60	0.02	
358192	8.15	19.23	45.00	50.20	16.00	0.11	High Fe and Zn content
1403557	8.12	18.21	26.20	44.80	18.80	0.15	
1558746	5.50	16.55	38.10	86.70	18.90	0.10	
2406044	7.19	18.20	20.30	71.80	18.70	0.16	
3585839	6.10	12.13	30.80	79.90	18.20	0.10	
3587319	7.90	19.21	42.40	69.80	19.20	0.05	
3827755	5.18	11.12	27.10	87.00	15.10	0.15	
4755489	6.00	11.12	43.40	81.20	16.00	0.20	
41868	3.15	7.15	41.70	16.60	12.20	0.21	High Cd content
295261	4.26	10.22	19.90	44.60	13.10	0.26	
3586080	8.99	12.29	43.00	88.80	17.00	0.25	
4097301	6.12	13.19	27.10	44.20	19.10	0.22	
3592850	7.15	13.15	29.10	59.20	13.20	0.26	
4319277	7.19	12.18	31.70	60.60	16.20	0.23	
3617481	8.19	15.12	15.60	45.30	17.60	0.21	
4755706	6.22	12.15	34.70	37.80	14.20	0.21	
4756035	6.19	14.85	41.10	51.20	19.20	0.22	
42274	6.19	11.11	46.30	45.20	17.20	0.21	
4970584	5.32	12.50	35.30	55.40	13.20	0.27	
346047	4.92	12.92	32.10	65.40	15.00	0.25	
1493157	6.30	12.92	31.30	50.20	17.60	0.21	
1987914	6.95	13.65	32.90	49.80	12.30	0.23	
766786	8.00	10.22	28.20	43.20	17.60	0.26	
778966	6.99	18.80	26.30	63.10	15.00	0.24	
2478018	6.33	12.18	27.50	43.70	17.40	0.26	
3686320	6.11	13.22	21.90	40.10	19.80	0.25	

**Table 3 genes-13-01052-t003:** Selected significant SNPs associated with some metals. Each SNP is detailed according to its chromosomal (Chr) position in megabase pairs (Mbp), significance value (–log10(*P*) value), explained phenotypic variance (R^2^), the effect of each allele, and its gene name in the first and second version of the wheat reference sequence (RefSeq), RefSeq v1.0 and RefSeq v2.1, respectively, announced by the International Wheat Genome Sequencing Consortium (IWGSC).

Metal	SNP	Chr	Position (Mbp)	-LOG10(P)	R^2^	Allele (Alternate)	Effect	Gene RefSeq v1.0	Gene RefSeq v2.1
Fe	IACX2594	5B	708	3.8	0.077	A (G)	2.09	TraesCS5B02G560400	TraesCS5B03G1356900
Mn				3.7	0.075	A (G)	0.96		
Fe	RAC875_rep_c106589_184			3.8	0.077	C (T)	2.09		
Mn				3.7	0.075	C (T)	0.96		
Fe	RAC875_rep_c106589_650			3.7	0.076	A (G)	−2.06		
Mn				4.0	0.082	A (G)	−0.99		
Fe	wsnp_Ex_c24031_33277293			3.8	0.077	A (G)	−2.06		
Mn				3.7	0.075	A (G)	−0.96		
Fe	wsnp_Ex_c24031_33277856			3.8	0.079	A (C)	2.04		
Mn				4.0	0.082	A (C)	0.98		
Cd	wsnp_Ex_c34597_42879693	6A	600	3.6	0.073	C (T)	−0.05	TraesCS6A02G375600	TraesCS6A03G0953900
Cu				3.1	0.060	C (T)	−7.75		
Fe				2.7	0.050	C (T)	−1.85		
Cd	wsnp_Ex_c2236_4189774			3.6	0.072	A (G)	−0.05		
Cu				3.1	0.061	A (G)	−7.81		
Fe				2.6	0.048	A (G)	−1.82		
Ni	BobWhite_c6300_169	3A	576	6.7	0.146	A (C)	−1.45	TraesCS3A02G331500	TraesCS3A03G0794000
Ni	RAC875_c60753_129		577	6.7	0.146	A (C)	14.48	NA	NA
Ni	RFL_Contig4431_279		577	6.6	0.146	A (C)	14.47	TraesCS3A02G331900	TraesCS3A03G0794600
Ni	Kukri_c5615_1214		579	4.4	0.090	A (C)	−8.09	TraesCS3A02G334100	TraesCS3A03G0799100
Ni	RAC875_c2140_128		581	4.4	0.090	A (C)	−8.08	TraesCS3A02G334700	TraesCS3A03G0801000
Ni	RAC875_c2140_103			4.4	0.090	A (C)	8.07		
Ni	Kukri_c37815_53		578	3.6	0.071	A (C)	5.87	TraesCS3A02G333100	TraesCS3A03G0796700

## Data Availability

Data of the measured metals is available upon request.

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
