# Peer review of "Genome-Wide Association Mapping of Grain Metal Accumulation in Wheat"

_genes, 2022, doi:10.3390/genes13061052_

Round 1

Reviewer 1 Report

·        -  The introduction: the author devotes large part of the introduction to explain the difference among QTL traditional studies, association studies (GWAM) and small part on genomic predictions. These methods are standard and widely used I do not see how this extensive explanation contributes to this study. I recommend shortening this part.

·         - Genomic prediction accuracy is highly depending on environment, heritability of the studied traits, and training populations. These points were not covered in this study. Why heritability was not calculated? Consequently, using genome predication in the title is not relevant in my opinion.

·        -  It was not clear why this 200 spring wheat lines were used and why it is important? Do we expect the same results in winter wheat? Why this study is different from what have been done before?

·         - Material and Methods: does the method and the applied concentrations reflects grower’s conditions? I missed the controls/reference lines also!

·        -  3 replicates within the same environment! So the conclusion on yield and GP is limited to this one environment?

·         - Table 1: it is good to add safety level to make it is easy to compare for the reader

·        -  Results: some of the five genotypes/lines selected for high yields showed high Cd level such as 393392 and 346403 (0.16, 0.21, respectively).

o    Related question is that the accepted safely threshold for Cd is 0.2 mg kg-1, seeing the distribution of this trait in the studied panel (0.003 to 0.266 mg kg-1) almost all the genotypes has acceptable level, do you think this panel was the right choice to study Cd accumulation in wheat as it seems optimized panel for breeding target and less suitable for genetic studies purpose!

·         - Figure 1: Fe seems to have some outliers, did the author check with the identified QTLs what would explain the few genotypes with very high Fe level (30 mg kg-1) and it is not continues distribution!

·        - Figure 1: for Zn it seems to have normal distribution and reflect additive effect background, however in Figure 3 we see major QTL on 3A and small one on 1A (need to be validated), any explanation?

·        - Table 3: title “Significant SNPs associated with the metals.” However, this table highlights only part of the identified QTLs not all, e.g: Zn QTL, Fe QTLs on 1A and 1B, why is that? Or maybe adjust the title to fit the purpose of this table.

·       -  3A chromosome contains major QTL for Zn but it is not reported in table 3 or in the supplementary table!!

·        -  Discussion: the author reports positive correlation between (Mn, Fe, Cu, and Zn, line 201) and in line 204 another positive correlation between (Cd, Mn, Fe, and Zn). I am confused between Cu and Cd!! Do we conclude positive corelation among all of them? Based on Figure 2, all was positive correlation however the significancy is low!

·         - Line 224: 2B chromosome SNPs associated with Fe and Zn, however based on Manhatten plots there is no QTL on 2B for Zn!

·         - In several examples one identified QTL in this study was identified in previous study linked to more metals (lines 228 till 230 as example). Why was it linked to more metals and only one detected in this study although, in most cases, the same metals are studied!!

Author Response

  • The introduction: the author devotes large part of the introduction to explain the difference among QTL traditional studies, association studies (GWAM) and small part on genomic predictions. These methods are standard and widely used I do not see how this extensive explanation contributes to this study. I recommend shortening this part.
    • Thank you for your comment. We have shortened this part as recommended.

  • Genomic prediction accuracy is highly depending on environment, heritability of the studied traits, and training populations. These points were not covered in this study. Why heritability was not calculated? Consequently, using genome predication in the title is not relevant in my opinion.
    • Due to the high cost, we measured the 6 metals with one replicate that represents a mix of three replicated seed samples. For each metal, we had 5 replications of one line that showed reasonably good similarity. As we agree with the reviewer's suggestion, we removed the prediction from the title. As the second reviewer suggested more modifications for the title, we have written a shorter and straightforward title “Genome-wide association mapping of grain metal accumulation in wheat”

  • It was not clear why this 200 spring wheat lines were used and why it is important? Do we expect the same results in winter wheat? Why this study is different from what have been done before?
    • We already showed a kind of similarity between our results and what has been introduced in winter wheat. The different point of our study is the analysis of the 6 metals together, so we can better explain the crosstalk between those metals. In addition, we confirmed some of the earlier marker-metal associations and introduced new ones.  

  • Material and Methods: does the method and the applied concentrations reflects grower’s conditions? I missed the controls/reference lines also!
    • We used seeds collected from the well-watered, normal, treatment as described in Saied et al 2022. We had no reference lines, as we were looking for the variation within the population itself.

  • 3 replicates within the same environment! So the conclusion on yield and GP is limited to this one environment?
    • As we mentioned before, we had one replicate that represent seeds from three different samples that were grown under the well-watered conditions in Egypt, as described in the recent article (Said et al 2022). However, we would expect some of the associated SNPs to be with main effects in many environments.

  • Table 1: it is good to add safety level to make it is easy to compare for the reader
    • Thank you for this valuable comment. We added the accepted threshold.

  • Results: some of the five genotypes/lines selected for high yields showed high Cd level such as 393392 and 346403 (0.16, 0.21, respectively).
    • The Cd level in genotype 393392 is below the accepted threshold “0.2”. In the case of genotypes 346403, we don’t think that the 0.01 above the threshold is toxic.

  • Related question is that the accepted safely threshold for Cd is 0.2 mg kg-1, seeing the distribution of this trait in the studied panel (0.003 to 0.266 mg kg-1) almost all the genotypes has acceptable level, do you think this panel was the right choice to study Cd accumulation in wheat as it seems optimized panel for breeding target and less suitable for genetic studies purpose!
    • Although the Cd content in the majority of the genotypes was below the threshold, we still see a huge variation in Cd accumulation which is good for mapping associated SNPs and candidate genes. The same applied to GY in our earlier study (Said et al 2022), as we selected five genotypes with high yield. In addition, we observed the possible variations for GY and Cd, low GY and high Cd, low GY and low Cd, high GY and high Cd, and high GY and low Cd. I believe that we wouldn’t be able to know without evaluation.
  • Figure 1: Fe seems to have some outliers, did the author check with the identified QTLs what would explain the few genotypes with very high Fe level (30 mg kg-1) and it is not continues distribution!
    • I believe there is a kind of confusion due to the ordering we used, top-down. We replaced this figure with the right order, left-right. Now, I believe the respected reviewer meant Ni, not Fe, as Fe is normally distributed, with 2 genotypes showing the lowest concentration and 5 genotypes showing the highest concentration. In the case of Ni, two genotypes showed 30 mg kg-1. We analysed the data with and without those two genotypes, the latter assuming they were outliers. Two QTL on chromosomes 1A and 3A were significant, -log10 (P) = 3.4 and 6.6, respectively, when those 2 genotypes were included. Excluding both genotypes revealed a less significant level, -log10 (P) = 3.5 and 4.4, respectively, for both QTL, however, 10 more SNPs were associated with Ni. To avoid false positives, we chose to report the stable QTL on 1A and 3A.

  • Figure 1: for Zn it seems to have normal distribution and reflect additive effect background, however in Figure 3 we see major QTL on 3A and small one on 1A (need to be validated), any explanation?
    • I believe that there is another confusion due to the different numbering. Now in the new figure, the 2 QTL on 3A and 1A were mapped for Ni, not for Zn.

  • Table 3: title “Significant SNPs associated with the metals.” However, this table highlights only part of the identified QTLs not all, e.g: Zn QTL, Fe QTLs on 1A and 1B, why is that? Or maybe adjust the title to fit the purpose of this table.
    • Thanks a lot for this comment. It would be difficult to present the 138 significant SNPs. Therefore, we selected some of them in table 3. We added, “Selected significant SNPs associated with the metals.”

  • 3A chromosome contains major QTL for Zn but it is not reported in table 3 or in the supplementary table!!
    • I believe that there is a confusion with figure 3 of the Manhattan plots. I might have numbered them in a wrong way, but it is Ni not Zn.

  • Discussion: the author reports positive correlation between (Mn, Fe, Cu, and Zn, line 201) and in line 204 another positive correlation between (Cd, Mn, Fe, and Zn). I am confused between Cu and Cd!! Do we conclude positive corelation among all of them? Based on Figure 2, all was positive correlation however the significancy is low!
    • That’s true; all metals were positively correlated, with different significant levels. However, Ni and Cd were not correlated. As we mentioned the correlation in a different context, we didn’t say that all traits were positively correlated.

  • Line 224: 2B chromosome SNPs associated with Fe and Zn, however based on Manhatten plots there is no QTL on 2B for Zn!
    • Thank you very much for your rigorous review. We really appreciate your effort. We have corrected the sentence.

  • In several examples one identified QTL in this study was identified in previous study linked to more metals (lines 228 till 230 as example). Why was it linked to more metals and only one detected in this study although, in most cases, the same metals are studied!!
    • The co-location indicates possible pleiotropy or linkage effects. We have reported several examples in table 3 and supplementary table 1.
      Several factors affect mapping QTL such as the metal variation within the populations, population structure, and using strict thresholds which might lead to reporting false-positive associations, or false negatives. Therefore, we might find all possible co-location with earlier studies, which is never common, or find part of the colocation as we reported here. In addition, some QTL are sensitive to the environments and are mapped in one environment but not others.
      In conclusion, we believe that the overlapping with earlier studies indicates growth QTL, otherwise dependent on the environment or population.

Reviewer 2 Report

The genetic factor in the context uptake of metals (especially heavy metals) is very important. Therefore, I assess the efforts of researchers to identify as many as 200 spring wheat varieties in this respect very positively. However, for this to happen, it needs to be slightly refined.

The wording in the title "desirable and undiserable" does not seem to be the best. In fact, the title can be accepted without them.

There is a lot of incorrect description of the results obtained in the work. For example the lines 119 to 123 should be amended. It  cannot to be describe the results in a scientific work by listing them one after another. The same applies to the paragraph that begins with line 138.

Also in the chapter discussion there is too much mention and too little real discussion.

The conclusions are inappropriate and should be redrafted.

Author Response

  • The genetic factor in the context uptake of metals (especially heavy metals) is very important. Therefore, I assess the efforts of researchers to identify as many as 200 spring wheat varieties in this respect very positively. However, for this to happen, it needs to be slightly refined.
    • Thank you very much for your positive assessment

  • The wording in the title "desirable and undiserable" does not seem to be the best. In fact, the title can be accepted without them.
    • We have re-written the title to be “Genome-wide association mapping of grain metal accumulation in wheat”

  • There is a lot of incorrect description of the results obtained in the work. For example the lines 119 to 123 should be amended. It  cannot to be describe the results in a scientific work by listing them one after another. The same applies to the paragraph that begins with line 138.
    • Thank you for your comment. I don’t see what is wrong with describing the data in that way. Maybe I’ve missed the sentences the reviewer was referring to, so I copied them below.
      • “A wide range of genetic variations was observed for all traits (Table 1). The lowest observed variation was for Mn, ranging from 3.0 to 9.9 mg kg-1 with an average of 6.1 mg kg-1. Zinc had the highest variation ranging from 9.9 to 88.8 mg kg-1. Iron and Ni showed moderate variations with minimum values of 5.3 and 11.1 mg kg-1, respectively, and maximum values of 19.2 and 31.0 mg kg-1, respectively. Frequency distribution (Figure 1) showed normal distribution for all traits”.
      • The correlation plot (Figure 2) showed a positive correlation between all measured traits, except Ni and Cd. The highest correlation was 0.70 between Mn-Fe, followed by 0.40 between Fe-Cu. The correlation between Fe and Cd was 0.32, and Cu and Cd was 0.23. The lowest correlation was 0.13 and was observed between Cu and Ni. and between Zn and Cd.
      •  
    • Also in the chapter discussion there is too much mention and too little real discussion.
      • We have amended the discussion

    • The conclusions are inappropriate and should be redrafted.
      • Thank you for your valuable comment. We have amended the conclusion to better reflect the work and findings.